# Grass Carp Reovirus Major Outer Capsid Protein VP4 Interacts with RNA Sensor RIG-I to Suppress Interferon Response

**DOI:** 10.3390/biom10040560

**Published:** 2020-04-06

**Authors:** Hang Su, Chengjian Fan, Zhiwei Liao, Chunrong Yang, Jihong Liu Clarke, Yongan Zhang, Jianguo Su

**Affiliations:** 1Department of Aquatic Animal Medicine, College of Fisheries, Huazhong Agricultural University, Wuhan 430070, China; suh_0210@163.com (H.S.); fanchengjian2016@163.com (C.F.); liaozhiwei1991@163.com (Z.L.); yonganzhang@mail.hzau.edu.cn (Y.Z.); 2Laboratory for Marine Biology and Biotechnology, Pilot Qingdao National Laboratory for Marine Science and Technology, Qingdao 266237, China; 3Norwegian Institute for Bioeconomy Research, 1430 Ås, Norway; jihong.liu-clarke@nibio.no; 4College of Veterinary Medicine, Huazhong Agricultural University, Wuhan,430070, China; chryang@mail.hzau.edu.cn

**Keywords:** grass carp reovirus (GCRV), major outer capsid protein VP4, molecular function, host/pathogen protein interaction, RIG-I-like receptor signaling pathway, immune evasion

## Abstract

Diseases caused by viruses threaten the production industry and food safety of aquaculture which is a great animal protein source. Grass carp reovirus (GCRV) has caused tremendous loss, and the molecular function of viral proteins during infection needs further research, as for most aquatic viruses. In this study, interaction between GCRV major outer capsid protein VP4 and RIG-I, a critical viral RNA sensor, was screened out by GST pull-down, endogenous immunoprecipitation and subsequent LC-MS/MS, and then verified by co-IP and an advanced far-red fluorescence complementation system. VP4 was proved to bind to the CARD and RD domains of RIG-I and promoted K48-linked ubiquitination of RIG-I to degrade RIG-I. VP4 reduced mRNA and promoter activities of key genes of RLR pathway and sequential IFN production. As a consequence, antiviral effectors were suppressed and GCRV replication increased, resulting in intensified cytopathic effect. Furthermore, results of transcriptome sequencing of VP4 stably expressed CIK (*C. idella* kidney) cells indicated that VP4 activated the MyD88-dependent TLR pathway. Knockdown of VP4 obtained opposite effects. These results collectively revealed that VP4 interacts with RIG-I to restrain interferon response and assist GCRV invasion. This study lays the foundation for anti-dsRNA virus molecular function research in teleost and provides a novel insight into the strategy of immune evasion for aquatic virus.

## 1. Introduction

Aquaculture is a major global industry with a total annual production exceeding 80 million ton and estimated value of almost 230 billion US dollar (FAO, 2016). In the past several decades, the aquaculture industry has made impressive progress and constitutes high quality protein for the world population accounting for nearly 50% of the global food fish supply. However, diseases caused by various aquatic viruses pose great hazards to aquaculture industry and a threat to food safety. People increasingly pay attention to the quality and safety of aquatic products and environmental pollution. But the utilization of various chemical drugs to protect aquaculture animals from diseases has been questioned increasingly. The potential risk of drug resistance, allergic reactions, and poisoning reactions caused by drug residues have a serious impact on the environment, farmed animals, and consumption of cultured products. Since viral diseases cause great hazards to the aquaculture industry and threat to food safety, research, and in-depth understanding the infection mechanism of the viruses is pivotal for the prevention of outbreaks of the diseases, and disease management post viral infections. To study the infection mechanism is an efficient approach to investigate the viruses targeting the function of each viral protein. Consequently, it is of importance to investigate the function of viral proteins.

Grass carp (*Ctenopharyngodon idella*), a freshwater aquaculture fish with a broad distribution, is one of the most important farmed fishes in China and also Asia. The yield of grass carp production reached 5.5 million ton (FAO, 2017). Hemorrhage disease caused by grass carp reovirus (GCRV) results in severe epidemic outbreaks and tremendous mortality of up to 90% in grass carp fingerlings every year [1], but its infection mechanism is still uncertain. GCRV is a double-stranded RNA (dsRNA) virus, belonging to group C, *Aquareovirus* genus in the *Spinareovirinae* subfamily, *Reoviridae* family [2] and it is regarded as the most virulent virus in *Aquareovirus* [3]. Its genome consists of 11 segments (termed S1 to S11), encoding 13 proteins, including seven structural proteins and six non-structural proteins [4], encased in a multilayered icosahedral capsid shell [5]. GCRV is classified into three types, which are respectively represented by strain GCRV-873 (type I), GCRV-GD108 (type II), and GCRV104 (type III) [2]. Among the three types, GCRV type II (GCRV-II) is the currently prevalent type and closer to *Orthoreovirus* than other known species of *Aquareovirus* [6]. Ortho- and aquareoviruses share nine homologous proteins encoded by the 10 or 11 genome segments of aquareovirus [3] as well as highly similar particle structures [7]. Seven of nine homologous proteins are structural, which are assembled into virions, including outer shell encoded by segment 6 of GCRV-II, defined as VP4 [8]. VP4 is the major outer capsid protein [9]. The open reading frame (ORF) of segment 6 in GCRV-097, a GCRV-II, is 1950 bp in length encoding a protein of approximately 68.4 kDa.

Innate immunity is the first line of host defense. In response to invading pathogens, pattern recognition receptors (PRRs) sense pathogen-associated molecular patterns (PAMPs) [10]. RIG-I-like receptors (RLRs) are a family of cytoplasmic PRRs which sense viral PAMPs [11] which are represented by the RIG-I (retinoic acid–inducible gene I, also called DDX58) [12]. RIG-I contains three domains: two tandem caspase-associated and recruitment domains (CARDs) presenting in the N-terminal, a central DExD box helicase/ATPase domain (DExD/H) (consisting of two RecA-like helicase domains, Hel1 and Hel2, and an insert domain, Hel2i) and a C-terminal repressor domain (RD) [13]. When unactivated, there is an auto-inhibited conformation of RIG-I, the two CARDs link to each other in a head to tail manner while the second CARD contacts with the Hel2i domain, shielding the CARDs-CARDs interaction between RIG-I and IFN-β promoter stimulator-1 (IPS-1), thereby interdicting the signal transduction [14]. RIG-I mainly recognizes RNAs with 5′ PPP or short dsRNA of a wide variety of RNA or DNA viruses [11]. Upon viral infection, the RD domain recognizes the 5′-PPP extremity of the blunt-end base-paired RNA and the helicase domain binds to the sugar-phosphate backbone of duplexed RNA, resulting in the release of CARDs [11]. The CARDs of RIG-I physically interact with the CARD of IPS-1, the adaptor protein of RLRs, to activate the downstream signaling cascade [15]. For RNA virus, fish interferon (IFN) antiviral response initiates from the pattern recognition of virus-RLRs component [16]. Signaling from RLRs pathway transmits to IFN regulatory factors (IRFs) via IPS-1, mediator of IRF3 activation (MITA, also known as STING) and TANK-binding kinase 1 (TBK1) successively, inducing phosphorylation of IRFs. Phosphorylated IRFs translocate from the cytoplasm to the nucleus where they activate IFN gene transcription by binding to ISRE/IRFE motifs presented in IFN promoters [12].

It is widely reported that RIG-I is modified by K63-linked and K48-linked ubiquitination [17]. K63-linked ubiquitination mediated by TRIM25 in K172 residue in RIG-I-CARDs is indispensable for IPS-1 recruitment and K154, K164, and K172 residues of RIG-I-CARDs are critical for Riplet-mediated K63-linked ubiquitination and antiviral signal transduction of RIG-I [18]. The K48-conjugated ubiquitination chain delivers the substrates to the proteasomes for degradation. RNF125, an E2 ubiquitin-conjugating enzyme, mediates the degradation of RIG-I via the K48-linked ubiquitination [19] to guarantee the basal protein levels, which are crucial for subsequently rapid signal activation. In mammals, binding to the K63 ubiquitin chain in the CARDs domain is essential for activation of RIG-I [20]. The ability to bind the K63 polyubiquitin chain of grass carp RIG-I CARDs is similar to that in mammals [21]. 

To survive in the presence of active host immune defense response, pathogens have evolved plenty of strategies to evade and exploit host immune signaling events [22]. Herpes simplex virus 1 (HSV-1) tegument protein UL37 is sufficient to induce RIG-I deamidation to prevent RIG-I activation by viral dsRNA [23]. RNA viruses deploy various mechanisms to disrupt signal transduction downstream of PRRs [22]. An elegant example that is shared by multiple RNA viruses, including hepatitis C virus, picornavirus, and enterovirus, is to cleave the MAVS adaptor off the mitochondrion membrane with a viral NS4/NS5 protease, thereby shutting down IFN induction in response to viral infection [24,25]. Viruses have evolved diverse strategies to halt or hijack antiviral signaling downstream of RIG-I [17]. Nonetheless, there have been few studies to date regarding the evasion mechanisms of GCRV to interfere with fish IFN production [26].

In the present study, GCRV major outer capsid protein VP4 was found to localize to early endosome, lysosome and endoplasmic reticulum (ER). RIG-I was screened out to interact with VP4 by GST pull-down, endogenous immunoprecipitation and subsequent LC MS/MS, verified by co-immunoprecipitation (IP) and bimolecular fluorescence complementation (BiFC). Binding between VP4 and the N- and C-terminal domains of RIG-I and enhanced degradation of RIG-I caused by K48-linked ubiquitination led to negative regulation of VP4 targeting the RLR signaling pathway and RIG-I triggered IFN responses. Consequently, VP4 facilitated GCRV replication leading to more intensive cytopathic effect. Furthermore, glucose-regulated protein 78 (GRP78) was found to bind to VP4, leading to ER stress. These data reveal that VP4 protein interacts with RNA sensor RIG-I and promotes its degradation to prevent RIG-I from activating the downstream RLR pathway and IFN responses for viral evasion. The present study focuses on the function analysis of interaction between virus and host protein. It explores of the infection mechanism of dsRNA virus and lays the foundation for the further antiviral molecular function research in teleost and provides a novel insight into the strategy of immune evasion for aquatic virus.

## 2. Materials and Methods 

### 2.1. Cell Culture, Viral Infection, and Antibodies

CIK (*C. idella* kidney) and FHM (fathead minnow) cells were respectively cultured in DMEM and M199 supplemented with 10% FBS (Gibco), 100 U/mL penicillin (Sigma), and 100 μg/mL of streptomycin (Sigma). Cells were incubated at 28 °C with 5% CO_2_ humidified atmosphere. Stably expressed CIK cell lines were selected using G418 treatment. GCRV-II strain GCRV-097 was conserved in our lab. For viral infection, CIK cells were plated for 24 h in advance and then infected with GCRV-097 at a multiplicity of infection (MOI) of 1 as previously described [21].

Mouse polyclonal antibodies of IFN1, IFN3, and MyD88 were prepared and conserved by our lab. The anti-IRF7 rabbit polyclonal antiserum was previously prepared in our lab [21,27]. Anti-IRF3 rabbit polyclonal antiserum was previously prepared and presented by Prof. Yibing Zhang, Institute of Hydrobiology, Chinese Academy of Sciences, Wuhan, China [28]. These antibodies were produced according to the corresponding sequences in grass carp and tested by using western blotting analysis before experiments. The VP4 antibody was produced and tested by the authors. Anti-HA tag (ab18181) mouse monoclonal antibody, anti-Flag tag (ab125243) mouse monoclonal antibody and anti-β-Tubulin rabbit polyclonal antibody (ab6046) were purchased from Abcam. Anti-GFP mouse monoclonal antibody (AE012) and anti-GST mouse monoclonal antibody (AE001) were purchased from Abclonal. IRDye^®^ 800CW Donkey anti-rabbit-IgG (926-32213) and anti-mouse-IgG (H + L) (926-32212) secondary antibodies were purchased from LI-COR. Goat-anti-mouse Ig-HRP conjugate secondary Ab (A0216) was purchased from Beyotime.

### 2.2. Plasmid Construction

The plasmids pGEX-4T-1, pDsRed1-C1, and pCMV-eGFP were employed as for the construction of expression vectors with specific primers (Appendix A). For subcellular localization, full-length open reading frames (ORFs) of VP4 and GRP78 were amplified and digested with restriction enzymes. VP4 was ligated into pGEX-4T-1 and pCMV-eGFP. GRP78 was ligated into pDsRed1-C1 and pCMV-eGFP. With the same method, VP4-Flag and GRP78-HA were ligated into pCMV-eGFP to construct overexpression vectors. Overexpression plasmids RIG-I-HA and RIG-I-Flag were saved in our lab [21]. pDsRed1-C1-RAB5, pDsRed1-C1-RAB7, and pDsRed1-C1-LAMP2 for subcellular localization studies were also conserved in our lab [29,30]. For dual-luciferase reporter assays, the valid promoters (RIG-I, IPS-1, STING, TBK1, IRF3, IRF7, IFN1, IFN3, IFNγ2, and NF-kB1, respectively) were cloned into pGL3-basic luciferase reporter vector (Promega), which had been previously constructed in our lab [21,27]. HA-Ub, HA-Ub-K63O, and HA-Ub-K48O plasmids were provided by Prof. Hong-Bing Shu (Wuhan University, Wuhan, China).

### 2.3. Prokaryotic Expression and Preparation of VP4 Polyclonal Antiserum

For the preparation of anti-VP4 polyclonal antiserum and GST pull-down, the full length of the GCRV-097 VP4 gene (GenBank accession number MN136091) was amplified with corresponding primers (Appendix A) and cloned into pGEX-4T-1 vector. The plasmid pGEX-4T1-VP4 was transformed into the E. coli BL21 (DE3) pLysS for prokaryotic expression. The fusion protein was induced by isopropyl b-D-1-thiogalactopyranoside (IPTG) and purified by GST Bind Resin (Genscript) chromatography. The purified protein was applied to immunize BALB/c mice to acquire the polyclonal anti-VP4 antiserum. The specificity was tested by western blotting (WB) assay [27].

### 2.4. GST Pull-Down, Immunoprecipitation, and LC-MS/MS

GST pull-down and immunoprecipitation (IP) assay were performed to explore CIK cell proteins interacting with VP4. For GST pull-down, CIK cell proteins were extracted with Membrane and Cytosol Protein Extraction Kit (Beyotime). Two hundred microliters of GST-VP4 and 200 μL of CIK protein solutions (1 μg/μL, diluted in TBS) were incubated at 4 °C for 30 min and then GST-bind resin (20 μL) was added. After 6 h 4 °C incubation, the resin was washed with TBS thoroughly and eluted with elution buffer (10 mM reduced glutathione and 50 mM Tris-HCl, pH 8.0) and then analyzed using SDS-PAGE and subsequent silver staining.

For IP, CIK cells were infected with GCRV for 24 h and co-IP was carried out with GST Ab, GST-VP4 Ab or negative serum (NS) using a Co-immunoprecipitation Kit (Pierce). The eluant was analyzed by SDS-PAGE and subsequent silver staining.

After GST pull-down or IP, LC-MS/MS analysis was performed on a Q Exactive mass spectrometer (Thermo Scientific) by Shanghai Applied Protein Technology Co. Ltd. LC MS/MS spectra were searched using the MASCOT engine (Matrix Science) against the actinopterygii UniProt sequence database (http://www.uniprot.org/), Grass Carp Genome Database (GCGD) (http://bioinfo.ihb.ac.cn/gcgd/php/index.php) and a grass carp transcriptome database in NCBI SRA browser (Bioproject accession number: SRP049081) [31].

### 2.5. Western Blotting and Co-Immunoprecipitation Analysis

For Western blotting (WB) analysis, protein extracts were separated by 8%–12% SDS-PAGE gels and transferred onto nitrocellulose membranes (Millipore). The membranes were blocked in fresh 2% albumin from bovine serum (BSA) dissolved in TBST buffer at 4 °C overnight, then incubated with appropriate indicated primary Abs for 2 h at room temperature. They were then washed three times with TBST buffer and incubated with secondary Ab for 1 h at room temperature. After washing four times with TBST buffer, the nitrocellulose membranes were scanned and imaged by an Odyssey CLx Imaging System (LI-COR) or an ImageQuant (GE). The results were obtained from three independent experiments.

For co-immunoprecipitation (co-IP), CIK cells in 10 cm^2^ dishes were co-transfected with the indicated plasmids for 48 h. The cells were lysed in IP lysis buffer (20 mM Tris [pH 7.4], 150 mM NaCl, 1% Triton X-100, 1 mM EDTA, 1 mM Na_3_VO_4_, 0.5 mg/mL leupeptin, 2.5 mM sodium pyrophosphate) (Beyotime) added with 1 mM PMSF for 30 min on ice, and the cellular debris was removed by centrifugation at 12,000× *g* for 30 min at 4 °C. The supernatant was transferred to a fresh tube and incubated with 1 mg of Ab with gentle shaking overnight at 4 °C. Protein A + G Sepharose beads (Beyotime) (30 mL) were added to the mixture and incubated for 2 h at 4 °C. After centrifugation at 3000× *g* for 5 min, the beads were collected and washed four times with lysis buffer. Subsequently, the beads were suspended in 20 mL 2 × SDS loading buffer and denatured at 95 °C for 10 min, followed by WB detection.

For the ubiquitination detection, FHM cells were seeded in dishes and transfected with corresponding plasmids. At 48 h post-transfection, the cells were treated with MG 132 for another 6 h. The cells were then harvested for IP with anti-Flag Ab and IB with anti-HA and anti-Flag Ab, respectively. The experiments were repeated at least three times. The histogram exhibits the relative protein expression levels, which were quantified using ImageJ software.

### 2.6. Confocal Microscopy and Transmission Electron. Microscope

For confocal microscopy, FHM cells were transfected with the indicated plasmids for 16 h and plated in observation dishes for confocal microscopy. Subsequently, the cells were washed, fixed and stained as previously reported [32]. Finally, the samples were observed with a confocal microscope (SP8, Leica). 

To analyze the ultrastructure of ER, stably transfected CIK cells were fixed with 2.5% glutaraldehyde in 0.1 M phosphate buffer (pH 7.2) for over 24 h at 4 °C. Ultrathin sections were prepared as described previously [33]. Images were viewed on an HT-7700 transmission electron microscope (TEM) (Hitachi, Japan).

### 2.7. Far-Red mNeptune Based Bimolecular Fluorescence Complementation System

A Far-red mNeptune-based bimolecular fluorescence complementation (BiFC) system is used to determine whether two proteins are interactive based on reconstitution of two non-fluorescent fragments of a fluorescent protein [34]. In the present study, the mNeptune-based BiFC system was used to visualize the interaction between VP4 and RIG-I, VP4, and GRP78 in CIK cells. Briefly, for VP4 and RIG-I, the ORF sequences of VP4 and RIG-I were cloned from CIK cells and inserted into the pMN155 and pMC156 plasmids, respectively. The final plasmids were named pVP4-MN155 and pMC156-RIG-I, which contained the N-terminal of mNeptune (mNeptune amino acid 1-155, MN155) after the C-terminal of VP4 and C-terminal of mNeptune (mNeptune amino acid 156-244, MC156) before the N-terminal of RIG-I, respectively. The coding regions were connected by the linker sequence GGGGSGGGGS. Plasmids pVP4-MN155 and pMC156-RIG-I were then transfected into CIK cells alone or together into CIK cells as described above. mNeptune was observed by confocal microscopy. Red mNeptune BiFC signals were measured with excitation at 640/20 nm and emission at 685/40 nm. The interaction between VP4 and GRP78 was verified by a similar method.

### 2.8. Dual Luciferase Reporter Assay

CIK cells were seeded in 24-well plates for 24 h. Co-transfection was performed with corresponding overexpression plasmid, target promoter luciferase plasmid, and internal control reporter vector (pRL-TK). After 24 h transfection, cells were infected with GCRV or were left uninfected. 24 h post-challenge, cells were washed with PBS and lysed with passive lysis buffer (Promega) for 30 min. Luciferase activities were detected by a Dual-Luciferase Reporter Assay System (Promega). The luciferase reading was normalized against those in the pRL-TK levels and the relative light unit intensity was presented as the ratio of luciferase of firefly to renilla.

### 2.9. Transcriptome Analysis

Empty vector or VP4 stably transfected CIK cells were screened by G418 as previously reported [32]. RNA isolation, cDNA library construction and sequencing were performed by Majorbio Biotech Co., Ltd, Shanghai, China. Data analysis was achieved using i-Sanger (https://www.i-sanger.com/). BLAST search and annotation were carried out against the Kyoto Encyclopedia of Genes and Genomes (KEGG) database, gene ontology (GO), Clusters of Orthologous Groups (COG), Nonredundant protsin (NR), Swiss-Prot, and Pfam. Gene expression according to transcriptome was validated by RT-qPCR with specific primers (Appendix A).

### 2.10. GCRV Titer and Crystal Violet Staining

Samples were infected with GCRV strain 097 (MOI = 1) for 24 h and supernatants were serially diluted 10-fold and incubated with CIK cells in a flat 96-well plate to determine the 50% tissue culture infective dose (TCID_50_). Cells were incubated at 28 °C for 7 d. On day 7, the plates were examined for the presence of viral cytopathic effect under the microscope.

For crystal violet staining, CIK cells were seeded into 24-well plate overnight and transiently transfected with empty vector or VP4. 24 h post-transfection, cells were infected with GCRV or uninfected. At 24 h post-infection, cells were washed and fixed with 4% paraformaldehyde for 15 min at room temperature and stained with 0.05% (*wt/vol*) crystal violet (Sigma, USA) for 30 min, then washed with water and drained. Finally, the plates were photographed under a light box (Bio-Rad).

### 2.11. siRNA Mediated Knockdown

Transient knockdown of VP4 in CIK cells was achieved by transfection of siRNA targeting VP4 mRNA. Three siRNA sequences (si-VP4#1 (sense 5′–3′), GGUCACCGUAUUGUUACAUTT, si-VP4#2 (sense 5′-3′), GGGUUGGUCUGAAGAUAUATT, si-VP4#3 (sense 5′–3′), GCAGCGUGUUGACAAGCUUTT) targeting different regions of VP4 were synthesized by GenePharma (Jiangsu, China). CIK cells were transfected with siRNA using GP-siRNA-Mate Plus (GenePharma, China) and infected with GCRV for 24 h post-transfection for another 24 h. The silencing efficiencies of the siRNA candidates were then evaluated by qRT-PCR, comparing with that in the negative control siRNA (si-NC) provided by the supplier. A preliminary experiment indicated that si-VP4#1 possessed the best silencing efficiency at a final concentration of 100 nM in mRNA levels. For WB, VP4 overexpression CIK cell line was plated in 6-well plates and transfected with si-VP4#1 using GP-siRNA-Mate Plus (GenePharma, China). The subsequent knockdown experiments were performed with si-VP4#1. CIK cells were transfected with siRNA#1 for 24 h and infected with GCRV for another 24 h post-infection. The cells were lysed, and proteins were extracted for WB.

### 2.12. Quantitative Real Time PCR Assay

Total RNA isolation was prepared according to a previous report [35]. Quantitative real time PCR (qRT-PCR) was established in Roche LightCycler^®^ 480 system, and EF1α was employed as internal control gene for cDNA normalization. The qRT-PCR amplification was carried out in a total volume of 15 μL, containing 7.5 μL of BioEasy Master Mix (SYBR Green) (Hangzhou Bioer Technology Co., Ltd.), 3.1 μL of nuclease-free water, 4 μL of diluted cDNA (200 ng), and 0.2 μL of each gene specific primer (10 μM) (Appendix A). The data were analyzed as previously described [36].

### 2.13. Statistical Analysis

The data were analyzed using an unpaired, two-tailed Student’s *t*-test. P values below 0.05 were regarded as being significant for all analyses (* *p* ≤ 0.05; ** *p* ≤ 0.01).

## 3. Results

### 3.1. VP4 Localizes to Early Endosome, Lysosome and ER, but Not Late Endosome 

To investigate the infection mechanism of GCRV, subcellular localization of major outer capsid protein VP4 was researched by confocal microscopy. Indirect immunofluorescence (IFA) with VP4 anti-serum was also attempted to research the subcellular localization, but a false positive effect influenced the results due to the limited quality of antibody. We co-transfected FHM cells with VP4-eGFP fusion vector and respective organelle protein markers, followed by DAPI staining of the cell nucleus. As shown in Figure 1, appearances of yellow signals indicating overlapping of green and red were observed in cells co-transfected with VP4 and RAB5, LAMP2, and GRP78, which are the marker protein of early endosome, lysosome and ER respectively. These results demonstrated that VP4 localizes to early endosome, lysosome and ER, but not late endosome. Thus, it can be speculated that after entry into the host cell, the GCRV particle is taken into the early endosome and transferred to the lysosome. The GCRV particle is disintegrated in the lysosome and VP4 is released into cytoplasm.

### 3.2. VP4 Interacts with CARD and the RD Domain of RIG-I

To research the molecular action of VP4 during GCRV infection, GST pull-down, endogenous IP and subsequent LC-MS/MS were performed to investigate CIK proteins binding with VP4 (Figure 2). Prokaryotic expressed GST-VP4 fusion protein was purified by affinity chromatography (Figure 2A) and VP4 polyclonal antiserum was prepared with the purified VP4 protein (Figure 2B). For GST pull-down, CIK cell protein was extracted and incubated with VP4 protein for binding, while GST protein was used as control. GST resin was then added into the complex. After washing and elution, the eluted protein mixture was analyzed using LC-MS/MS (Figure 2C left). Meanwhile, CIK cells were infected with GCRV and IP with VP4 polyclonal antiserum, GST antibody (Ab) and negative serum, respectively, using an IP kit, followed by LC-MS/MS (Figure 2C right). Summarizing the LC-MS/MS results, a large number of interactive proteins were identified from the proteomic dataset, of which 61 candidate proteins were settled matching with at least two unique peptides (Appendix A). Among the candidate interactive proteins, RIG-I and GRP78 were obtained and to validate.

The direct interaction between VP4 and RIG-I was verified by co-IP (Figure 3A) and BiFC (Figure 3B). Positive bands were observed in both co-IP results conducted by HA-tag fused with RIG-I and flag-tag fused with VP4 (Figure 3A). According to the BiFC results, red fluorescence of mNeptune was not detected in single VP4 or RIG-I protein overexpressed cells. When CIK cells were transfected with both VP4 and RIG-I plasmids respectively linked with each terminal of the mNeptune protein, these two fusion proteins presented closely and sent out red fluorescence signal (Figure 3B). To further find out the specific interaction domain of RIG-I, co-IP was performed, and the results indicated that RIG-I-CARD and RIG-I-RD domain could interact with VP4 but not the helicase domain of RIG-I (Figure 3C). Therefore, the interaction between VP4 and RIG-I was proved, and the binding sites were found to be CARD and the RD domain of RIG-I

### 3.3. VP4 Inhibits RIG-I-Triggered IFN Response

Overexpression was utilized to measure the effect of VP4 on the RIG-I mediated signaling pathway because of the unstable reverse genetic system of GCRV and limited number of antibodies available to specific molecules in fish. Since RIG-I is a central PRR in antiviral signaling pathway, sensing PAMPs in cytoplasm, the interaction between VP4 and RIG-I is supposed to play a great role in GCRV immune evasion. As shown by dual-luciferase assay, various degrees of decreased promoter activities of RLR signaling pathway key molecules, including RIG-I, IPS-1, STING, TBK1, and IRF7, were detected after VP4 overexpression under GCRV infection or when uninfected (Figure 4A). A similar tendency was seen in results measured by qRT-PCR, but the mRNA level of IRF7 was significantly induced by VP4 (Figure 4B). In addition, IRF3 and IRF7 expression and phosphorylation were tested by Western blotting using corresponding antiserum. The WB results illustrated that protein expression and phosphorylation of IRF3 were reduced by VP4 during GCRV infection, but IRF7 presented a contrary condition (Figure 4C). As a consequence of the inhibition of the RLR signaling pathway, the mRNA expression levels of IFN1, IFN3 and IFNγ2 were found to be decreased by VP4 (Figure 4D). IFN1 and IFN3 production were decreased by VP4 (Figure 4E). When co-transfected with RIG-I, promoter activities and mRNA levels of IFN1, IFN3, and IFNγ2 were reduced compared with single RIG-I overexpression (Figure 4F,G) indicating that VP4 inhibited RIG-I triggered IFN responses. In addition, VP4 also restricted RIG-I induced NF-κB1 mRNA and promoter activity (Figure 4F,G). 

To understand the negative regulation mechanism of VP4 and whether VP4 can affect the K63- and K48-linked ubiquitination of RIG-I, *in vitro* ubiquitination assays were performed in FHM cells (Figure 4H). The results showed that VP4 overexpression enhances both K63- and K48-linked ubiquitination of RIG-I in a dose-dependent manner (Figure 4H). Additionally, the result of the RIG-I degradation assay demonstrated that overexpression of VP4 induces RIG-I degradation (Figure 4I). These results reveal the down-regulation mechanism of VP4 on RIG-I.

To further verify the regulation mechanism of VP4 on the host antiviral signaling pathway and find out why IRF7 was upregulated by VP4 on the mRNA level, transcriptome sequencing was performed (Figure 5). CIK cells stably expressing empty vector (CIK-vector) (control) or VP4 (CIK-VP4) were selected by G418 treatment and subjected to RNA sequencing (RNA-seq) after RNA extraction. The CIK-vector sample yielded 55.4 million clean reads and the CIK-VP4 sample yielded approximately 50.2 million clean reads (Appendix A). Furthermore, 56,445 transcripts were detected after assembly and the length distribution of all transcripts is shown in Appendix A. All reads have been submitted to the Sequence Read Archive at NCBI (Accession Number: SRP212372). All transcript sequences were searched in the Nonredundant protein (NR), Swiss-Prot, Pfam, Kyoto Encyclopedia of Genes and Genomes (KEGG), Clusters of Orthologous Groups (COG), and Gene Ontology (GO) databases. Totally, 32,749 genes were annotated (Appendix A) and the annotation histogram of KEGG is shown in Figure 5A. The KEGG functional enrichment of differentially expressed (fold-change of at least 4) genes (DEGs) analysis showed that focal adhesion, ECM-receptor interaction and the Pl3k-Akt signaling pathway were activated the most (Figure 5B), showing that after VP4 stimulation, these three signaling pathways had the strongest responses. Anti-virus and ER stress related genes were focused on, and the raw expression levels are shown in Figure 5C. To validate the RNA-seq results, representative genes were selected for validation by qRT-PCR and WB (Figure 5D–F). The RLR signaling pathway was inhibited by VP4, while IRF7 was induced (Figure 5D). NF-κB was also reduced and IκBα, a NF-κB inhibitor, was induced (Figure 5E). Interestingly, MyD88 was found to increase after VP4 stimulation (Figure 5F left) and MyD88 was promoted by VP4 in protein level (Figure 5F middle and right). This result demonstrated that VP4 activated MyD88 and MyD88 related TLR signaling pathway during GCRV infection, which is accountable for the increase of IRF7 upon VP4 overexpression.

### 3.4. VP4 Facilitates GCRV Replication

To determine whether VP4 interferes with the cellular IFN response to facilitate viral RNA replication, CIK cells were transfected with VP4 and stimulated with GCRV (Figure 6). Total RNAs were extracted and monitored by RT-qPCR. As shown in Figure 6A, mRNA expression levels of ISGs, such as myxovirus-resistant 2 (Mx2) and GCRV induced gene 1 (gig1), in cells overexpressing VP4 were reduced compared to the levels in control cells upon GCRV infection or when uninfected. After infection with GCRV, the viral VP1, VP56, NS38 and VP35, which are respectively encoded by GCRV-II segment S1, S7, S10, and S11, were up-regulated in cells overexpressing VP4 (Figure 6B). In addition, titration analysis indicated that VP4 upregulated the titer of GCRV in CIK cells transiently or stably transfecting VP4 (Figure 6C). Crystal violet staining also indicated that VP4 promotes the cell death caused by GCRV infection (Figure 6D). From these data, it can be demonstrated that VP4 interferes with ISG expressions and facilitates viral RNA synthesis, leading to increased GCRV titer and promoted cytopathic effect.

### 3.5. Knockdown of VP4 Potentiates GCRV-Induced IFN Responses

Knockdown of VP4 potentiates IFN responses and reduces GCRV replication and titer (Figure 7). si-VP4#1 was screened out to knockdown VP4 expression among three pieces of siRNAs by qRT-PCR and then verified by WB (Figure 7A). Knockdown of VP4 increased RIG-I mRNA expression (Figure 7B) and activated IFN responses (Figure 7C left). IFN1 production was also promoted by knockdown of VP4 (Figure 7C middle and right). ISGs were promoted (Figure 7D) and GCRV replication and titer were inhibited after knockdown of VP4 (Figure 7E,F). These results proved the conclusion obtained by VP4 overexpression, showing that VP4 inhibited RIG-I induced IFN response and enhanced GCRV replication.

### 3.6. VP4 Associated with GRP78 Leading to Unfolded Protein Response and ER Stress

GRP78 was obtained by LC MS/MS to interact with VP4 (Appendix A). Co-IPs and BiFC further proved the interaction (Appendix A). GRP78 is a ubiquitous molecule chaperone in ER stress [37]. TEM observation showed that VP4 caused morphological change of ER, tumidness, expansion and degranulation (Appendix A). In addition, activating transcription factor 4 (ATF4), a transcriptional activator in the PERK-eIF2α pathway, was significantly induced at mRNA level by VP4, while activating transcription factor 6 (ATF6) and TNF receptor-associated factor 2 (TRAF2) did not react under VP4 overexpression, showing that VP4 caused ER stress and PERK-eIF2α signaling pathway mediated unfolded protein response (Appendix A).

## 4. Discussion

Fish is a preeminent global animal protein source of great importance. To protect aquaculture production from diseases caused by viruses, it is indispensable to investigate the molecular function of viral proteins. In the present study, the function of GCRV major outer capsid protein VP4 and its regulatory mechanism were researched. Subcellular localization and interacting host protein of VP4 were investigated and found that VP4 directly bind to RIG-I, a significant cytoplasmic RNA sensor, to inhibit RIG-I triggered IFN responses and promote GCRV replication. We also elaborated the regulatory mechanism of downstream signaling pathway related to RLR and TLR signaling pathway. The present research stated the inhibition target of VP4 to restrain host innate antiviral response and promote GCRV replication and expounded the regulatory mechanism.

GCRV-I is the most widely studied type among the three types of GCRV. It is reported that caveolae/raft-mediated endocytosis is the primary entry pathway for GCRV-I [38] and disruption of clathrin-dependent trafficking results in failure of GCRV-I cellular entry [39]. Thus, the invasion mechanism of GCRV-I remains controversial. The identity of amino acid sequences is less than 20% between GCRV-II and GCRV-I. Therefore, GCRV-II may in all probability take a different approach to realize cell entry; however, few studies have been published. For decades, researchers have attempted to find out the specific receptor for GCRV-II to enter into the host cell, but an increasing number of reports indicate that the biological process is not performed through specific ligand–receptor interaction on cell membrane. In the present study, we focused on the major outer capsid protein of GCRV-II, VP4, to study its functions in GCRV-II infection and elucidate the infection mechanisms of GCRV. A majority of exogenous substances enter into cells by phagocytosis, macropinocytosis, clathrin-dependent endocytosis, caveolin-dependent endocytosis or clathrin- and caveolin-independent pathways. After entry into cells, particles are transferred to early endosome or lysosome [40]. In this study, we gave up the method of IFA because of the limited quality of VP4 antiserum, and the subcellular localization results by overexpression showed that after entry into cells, GCRV virions are taken into early endosome and lysosome for subsequent dissociation (Figure 1). Exogenous substances are transferred to lysosome via early endosome and late endosome under normal circumstances. However, the present results show that late endosome is not involved in the process of VP4 invasion suggesting that GCRV may infect host cells by two distinct approaches at the same time, so further research is needed. Then VP4 is separated from virions and released into cytoplasm. Additionally, in accordance with the transcriptome analysis in the present study, KEGG enrichment of DEGs (Fold = 4) demonstrates that the most enriched three signaling pathways activated by VP4 are the focal adhesion pathway, ECM receptor interaction pathway and PI3K-Akt signaling pathway (Figure 5B). From these results, it can be concluded that VP4 is closely related to the interaction with extracellular matrix (ECM) and receptor proteins and the adhesion of GCRV particles before entry into cells, suggesting that GCRV-II adheres onto cells by the VP4-ECM receptor complex before proceeding with the subsequent invasion. However, the specific mechanism still needs further studies.

To investigate the functions of major outer capsid proteins and infection mechanisms during virus infection, VP4 was expressed in *Escherichia coli* and purified, then GST pull-down and co-IP were performed (Figure 2). After release into cytoplasm, VP4 was found to directly interact with a cytoplasmic PRR, RIG-I, which senses viral RNAs (Figure 3). Zhao et al. have reported that herpes simplex virus 1 (HSV-1) tegument protein UL37 targets the helicase domain of RIG-I to deamidate RIG-I, preventing RNA-induced activation of innate immune signaling [23]. Endogenous molecule LRRC25 inhibits type I IFN signaling by targeting ISG15-associated RIG-I for autophagic degradation [41]. The present study discovered that GCRV VP4 targets the CARD and RD domain of RIG-I to reduce the innate antiviral response. When un-activated, RIG-I was auto-inhibited, and the interaction between VP4 and the CARD and RD domains of RIG-I could physically keep RIG-I remaining inactivated and could not recognize viral RNA, so that RIG-I could not trigger downstream IPS-1 and signal transduction.

Independent of GCRV infection, VP4 overexpression inhibits mRNA levels and promoter activities of RIG-I and its downstream adaptors and other key molecules (Figure 4A,B). Similar to mammals, fish IFN antiviral response is initiated through the pattern recognition of virus component by TLRs and RLRs [26]. IFN1 and IFN3 mRNA and protein production were reduced by VP4 (Figure 4D,E). Moreover, when co-treated with RIG-I, VP4 could silence RIG-I-mediated IFN responses (Figure 4F,G), which suggests that VP4 specifically targets RIG-I. Additionally, these results were ensured by siRNA-mediated knockdown of VP4 (Figure 7). Numerous studies have mentioned that almost all the host cellular processes are regulated by the ubiquitin system, which is important in regulation of protein stability, immune activation, and host-pathogen interplay by protein ubiquitination [11]. K48-conjugated ubiquitination leads to degradation of RIG-I, which presents an opposite effect to K63 on RLR pathway signal transduction. In the present study, both K63- and K48- linked ubiquitination of RIG-I were enhanced by VP4 (Figure 4H), but according to the degradation assay, RIG-I protein expression receded along with rising VP4 expression (Figure 4I). Thus, it can be seen that K48-linked proteasome degradation played a stronger role in the effect of VP4 on RIG-I. It was revealed that VP4 could lead to intensified degradation of RIG-I to prevent RIG-I from triggering innate immune responses resulting in weakened signaling transduction. Accordingly, VP4 performs two inhibition strategies on RIG-I, which are physical interaction to block the recognition of RNA and degradation of RIG-I leading to weaker signaling transduction.

As a consequence of the restrained IFN response and ISGs, GCRV replication recovered, mRNAs of GCRV segments were up-regulated, GCRV titer increased and cell death caused by GCRV infection was intensified (Figure 6). These results reveal that VP4 has a function as a negative factor targeting RIG-I to accomplish GCRV immune evasion. HSV-1 USP21 [42] and ubiquitin ligase RNF5 [43] regulate antiviral responses by mediating degradation of STING to achieve immune evasion. Along with the IFNs restrained by VP4, NF-κB1 was also suppressed by VP4 in mRNA and promoter activity levels (Figure 4D,F,G). Trans-localization of NF-κB from cytoplasm to nucleus induces expression of proinflammatory cytokines. The inhibited NF-κB1 by VP4 indicates that VP4 also affects the inflammatory response to inhibit host immunity via the RIG-I-mediated signaling pathway. The regulation network during virus infection is complicated and further research on each point is necessary.

Interestingly, after VP4 stimulation, IRF7 was upregulated in mRNA and promoter activity levels (Figure 4A,B). Furthermore, IRF7 expression and phosphorylation were enhanced (Figure 4C). The results hinted that VP4 probably activates other signaling pathways. As expected, the MyD88-dependent TLR signaling pathway was activated in VP4, stably expressing CIK cells according to transcriptome analysis (Figure 5F). These results illustrate that upon GCRV or other virus infection, pathogenic molecules target the host immune system to achieve immune evasion, but at the same time there exists an alternative host anti-pathogen signaling pathway to react against etiological agents. Along with the development of virus evolution, hosts also evolve corresponding strategies to fight against external risks.

ER is a significant organelle responsible for intracellular protein and lipid synthesis, xenobiotic detoxification and cellular calcium storage [44]. Once ER homeostasis is disrupted, unfolded and misfolded proteins accumulate in ER lumen, creating a condition which is defined as ER stress, activating unfolded protein response (UPR) [45]. GRP78 is an ER-resident chaperone, being enhanced when UPR triggers a unique signaling cascade from ER to nucleus [45]. Upon ER stress, GRP78 binds to unfolded proteins causing dissociation from three ER-transmembrane transducers of the respective UPR branches, PERK-eIF2α pathway, IRE1α-XBP1 pathway and ATF6 pathway [46]. Each pathway culminates in transcriptional regulation of gene expression and contributes to the overall maintenance of homeostasis in the ER during stress [47]. However, in fish, to our knowledge, there are very limited studies involving ER stress and UPR in response to virus infection [48,49]. As shown in Appendix A, VP4 interacts with GRP78. Three UPR branches were detected, ATF4, an eventful transcriptional factor in the PERK-eIF2α pathway, was activated but not ATF6 or TRAF2, indicating that VP4 triggers UPR through the PERK-eIF2α pathway. The ER stress and UPR caused by VP4 implies that GCRV infection leads to destruction of intracellular homeostasis. Thus, GCRV not only attacks the host immune system, but also affects the host cell homeostatic system.

## 5. Conclusions

Based on the results in the present study, we concluded that after released into cytoplasm, GCRV major outer capsid protein VP4 targets the cytoplasmic RNA sensor RIG-I to restrain RIG-I induced antiviral responses by structurally resistance and protein degradation of RIG-I via K48-linked ubiquitination. Furthermore, VP4 suppresses RIG-I regulated IFN response. Although VP4 activates other antiviral signals, it eventually inhibits IFN and cytokine production and promotes viral replication (Figure 8). The conclusion from this analysis reveals the molecular function of VP4 during GCRV infection and a competitive mechanism between virus evasion and antiviral immunity. The present study focuses on the function analysis of interaction between virus and host protein and explores of the infection mechanism of dsRNA virus. It lays the foundation for the further antiviral molecular function research in teleost and provides a novel insight into development of antiviral immunology research on viral escaping strategy and viral protein function research.

## Figures and Tables

**Figure 1 biomolecules-10-00560-f001:**
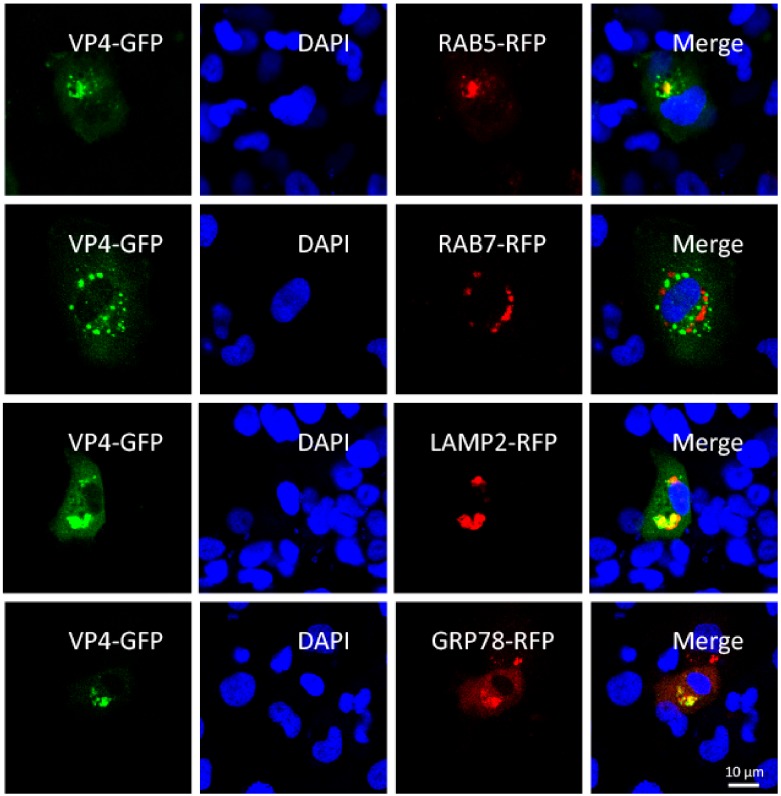
VP4 localizes to early endosome, lysosome, and endoplasmic reticulum (ER), but not late endosome. FHM cells were respectively transiently co-transfected with VP4-eGFP and LAMP2, a lysosome protein marker, VP4-eGFP and RAB5-RFP, an early endosome protein marker, VP4-eGFP and RAB7-RFP, a late endosome protein marker or VP4-eGFP and GRP78, an ER protein marker, and seeded on observation dishes for confocal microscope examination. After 48 h, the cells were fixed with 4% (*v*/*v*) paraformaldehyde and stained with DAPI. All samples were subsequently visualized using a confocal microscope. Green signals represent overexpressed VP4 and red signals stand for overexpressed LAMP2, RAB5, RAB7, or GRP78. The blue staining indicates nucleus. The yellow signals in the merged images indicate the co-localization between VP4 and organelle (original magnification × 40). All the experiments were repeated independently at least three times.

**Figure 2 biomolecules-10-00560-f002:**
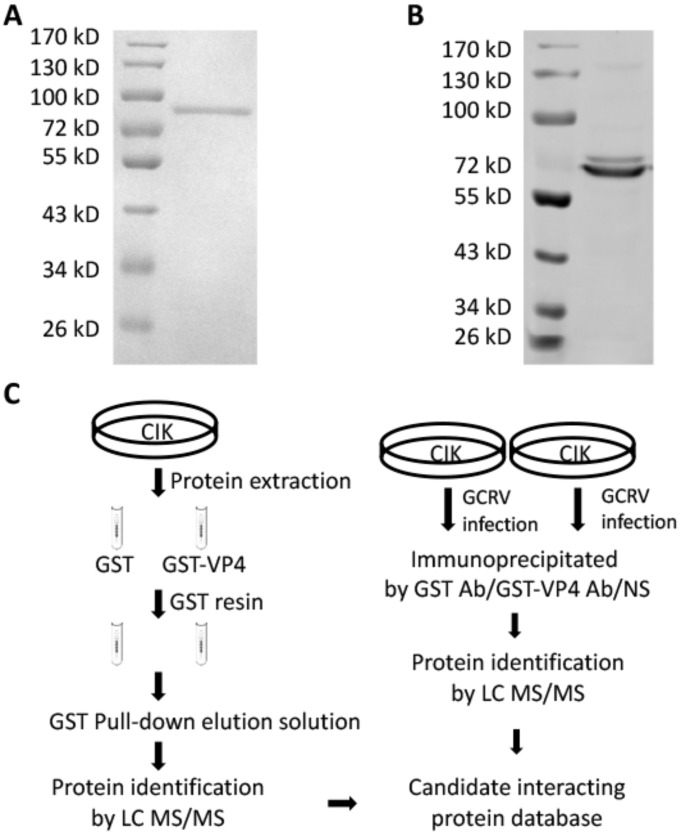
GST pull-down and co-immunoprecipitation (co-IP). (**A**) Purified prokaryotic GST-VP4 recombinant protein was analyzed by SDS-PAGE. Lane 1: Marker; Lane 2: GST-VP4. (**B**) GST-VP4 polyclonal antiserum prepared by our lab was detected by WB using CIK (*C. idella* kidney) cell sample infected with grass carp reovirus (GCRV). Lane 1: Marker; Lane 2: CIK protein detected by GST-VP4 antiserum. (**C**) Strategy of GST pull-down and co-IP. Left: Strategy for analyzing the interacting proteins of VP4 via GST pull-down and LC-MS/MS. CIK whole proteins were exacted from CIK cell line and incubated with prokaryotic GST or GST-VP4 protein. Pull-down was carried out with GST resin and the elution was analyzed by LC-MS/MS analysis on a Q Exactive mass spectrometer. Right: Strategy for analyzing the interacting proteins of VP4 via co-IP and LC-MS/MS. CIK cells were infected with GCRV for 24 h and co-IP with GST monoclonal Ab, GST-VP4 polyclonal antiserum or negative serum (NS). LC-MS/MS analysis was performed on a Q Exactive mass spectrometer. One experiment representative of three independent experiments was performed with three biological replicates.

**Figure 3 biomolecules-10-00560-f003:**
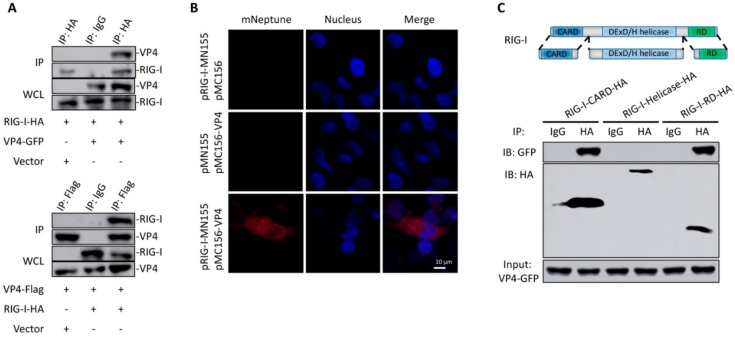
Identification of RIG-I as an interacting protein of VP4. (**A**,**B**) Verification of the interaction between VP4 and RIG-I via co-IP assay and bimolecular fluorescence complementation assay (BiFC). (A) Upper: FHM cells were co-transfected with VP4-eGFP/vector and RIG-I-HA for 48 h. Co-IP was performed with anti-HA monoclonal Ab and mouse IgG (control), and immunoblotted (IB) with the respective Abs. Below: FHM cells were co-transfected with RIG-I-HA/vector and VP4-Flag for 48 h. Co-IP was performed with anti-Flag monoclonal Ab and mouse IgG (control), and immunoblotted (IB) with the respective Abs. (**B**) Imaging of the VP4-RIG-I interaction by using far-red mNeptune-based BiFC in vivo. pRIG-I-MN155 and pMC156-VP4 were transfected alone or co-transfected into CIK cells under normal conditions. In the BiFC system, the fluorescence of the mNeptune channel was red and the nucleus was stained with DAPI. The images were acquired using confocal microscopy under a 40 × objective lens. Appearance of red fluorescence represents the positive observation. The BiFC experiments were repeated three times and the images were selected and cropped to show the positive reactions clearly. (**C**) Upper: Schematic representations of full-length RIG-I and the three domains constructed in the present study. Below: VP4 interacts with RIG-I-CARDs and RIG-I-RD, but not RIG-I-DExD/H Helicase domain. FHM cells were co-transfected with 4 µg VP4-eGFP and 4 µg RIG-I-CARD-HA or RIG-I-Helicase-HA or RIG-I-RD-HA for 24 h in 10 cm^2^ dishes. Co-IP was performed using anti-HA Ab, and mouse IgG was used as control. IPs were analyzed by IBs with anti-HA and anti-GFP, respectively. Expression of VP4-eGFP (input) was examined with anti-GFP Ab. All the co-IP and BiFC assay were repeated independently at least three times.

**Figure 4 biomolecules-10-00560-f004:**
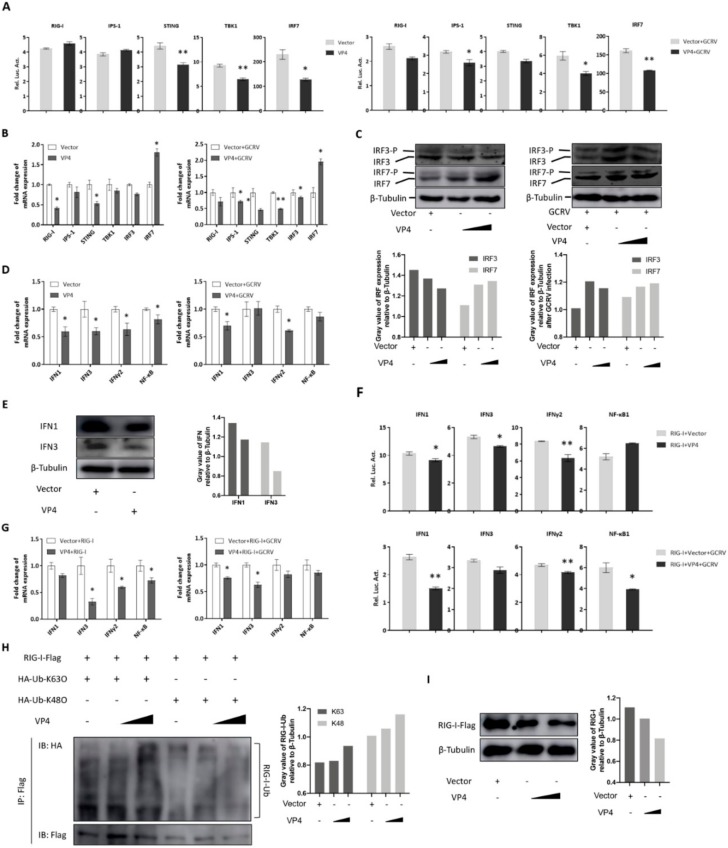
VP4 associated with RIG-I inhibits RLR triggered-IFN responses. (**A**) VP4 overexpression inhibits RLR signaling pathway key gene promoter activities. CIK cells seeded in 24-well plates overnight were co-transfected with 380 ng of VP4/empty vector, 380 ng of each target plasmid (pRIG-Ipro-Luc, pIPS-1pro-Luc, pSTINGpro-Luc, pTBK1pro-Luc and pIRF7pro-Luc) and 38 ng of pRL-TK. 24 h later, the cells were infected with GCRV or uninfected. The luciferase activities were examined at 24 h post-challenge. (**B**) VP4 overexpression decreases RLR related gene mRNA expressions in uninfected (left) or GCRV infected (right) CIK cells. CIK cells transiently transfected with VP4/empty vector were seeded in 12-well plates. After 24 h, CIK cells were uninfected (left) or infected with GCRV (right). 24 h post-infection, total RNA was extracted and examined for RIG-I, IPS-1, STING, TBK1 and IRF7 mRNA expression. (**C**) VP4 reduces IRF3 but induces IRF7 protein expression and phosphorylation. CIK cells transiently transfected with VP4/empty vector were seeded in 6-well plates. After 24 h, CIK cells were uninfected (left) or infected with GCRV (right). 24 h post-infection, cell lysate was used for WB analysis using IRF3/IRF7 polyclonal antiserum. β-Tubulin was used to normalize the protein concentration. All the experiments were repeated independently at least three times. The histograms below the western blotting results exhibit the relative expression levels, which were quantified using ImageJ software. (**D**) VP4 overexpression decreases IFN1, IFN3, IFNγ2 and NF-κB1 mRNA expression in uninfected (left) or GCRV infected (right) CIK cells. CIK cells transiently transfected with VP4/empty vector were seeded in 12-well plates. After 24 h, CIK cells were uninfected (left) or infected with GCRV (right). 24 h post-infection, total RNA was extracted and examined for IFN1, IFN3, IFNγ2, and NF-κB1 mRNA expression. (**E**) VP4 reduces IFN1 and IFN3 production. CIK cells transiently transfected with VP4/empty vector were seeded in 6-well plates. After 24 h, cell lysate was used for WB analysis using IFN1 or IFN3 polyclonal antiserum. All the experiments were repeated at least three times. The histogram exhibits the relative protein expression levels, which were quantified using ImageJ software. (**F**) VP4 inhibits RIG-I-triggered IFN1, IFN3, IFNγ2, and NF-κB1 promoter activities. CIK cells seeded in 24-well plates overnight were co-transfected with 380 ng RIG-I and 380 ng of VP4-eGFP/empty vector, 380 ng of each target plasmid (pIFN1pro-Luc, pIFN3pro-Luc, pIFNγ2pro-Luc, and pNF-κB1pro-Luc) and 38 ng of pRL-TK. Twenty-four hours later, the cells were infected with GCRV or left uninfected. The luciferase activities were examined at 24 h post-challenge. (**G**) VP4 overexpression blocks RIG-I-triggered IFN1, IFN3, IFNγ2, and NF-κB1 mRNA expression in uninfected (left) or GCRV infected (right) CIK cells. CIK cells transiently transfected with RIG-I and VP4 or empty vector were seeded in 12-well plates. After 24 h, CIK cells were uninfected (left) or infected with GCRV (right). 24 h post-infection, total RNA was extracted and examined for IFN1, IFN3, IFNγ2, and NF-κB1 mRNA expression. (**H**) VP4 enhances K63- and K48-linked ubiquitination of RIG-I. FHM cells were seeded in 10 cm^2^ dishes for 24 h and transfected with 1 µg HA-Ub-K63O or HA-Ub-K48O, VP4 (0, 1.5, and 3 µg) together with decreasing amounts of empty vector (3, 1.5, and 0 µg), 4 µg RIG-I-Flag. At 48 h post-transfection, the cells were treated with MG132 for 6 h. The cells were then harvested for IP with anti-Flag Ab and IB with anti-HA and anti-Flag Ab, respectively. All the experiments were repeated at least three times. The histogram exhibits the relative protein expression levels, which were quantified using ImageJ software. (I) VP4 suppresses RIG-I production. FHM cells were transfected with 2 µg RIG-I-Flag, VP4 (0, 1, and 2 µg) together with decreasing amounts of empty vector (2, 1, and 0 µg). At 48 h post-transfection, the cells were harvested for IB with anti-Flag Ab and β-Tubulin Ab. All the experiments were repeated at least three times. The histogram exhibits the relative protein expression levels, which were quantified using ImageJ software. Data of reporter assays and qPCR are shown as mean ± SD of 4 wells of cell per group and are from one experiment representative of three independent experiments. Significance was calculated in relation to the control group. * *p* < 0.05, ** *p* < 0.01 (two tailed Student’s tests). The relative transcription levels were normalized to the transcription level of EF1α gene and are represented as fold induction relative to the transcription level in control cells, which was set to 1.

**Figure 5 biomolecules-10-00560-f005:**
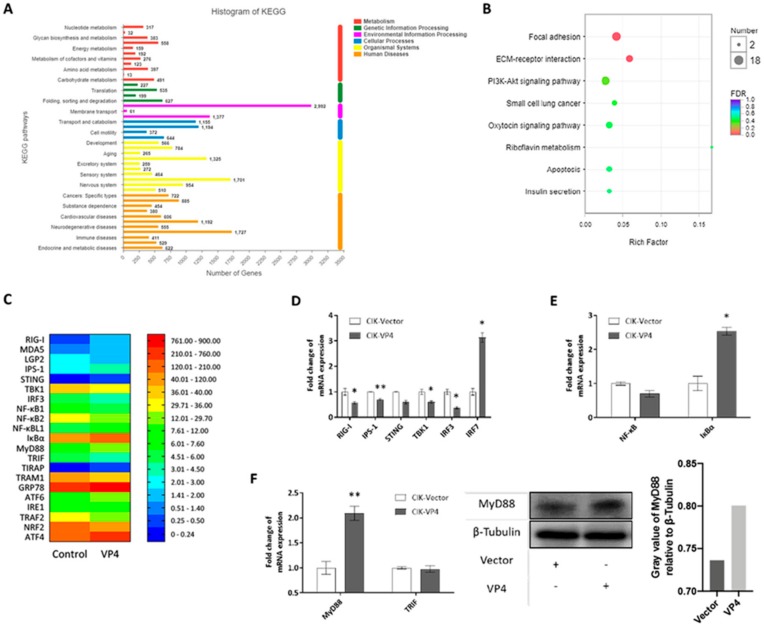
Transcriptome analysis of VP4 overexpressed in CIK cells. (**A**) Functional annotation of sequences based on Kyoto Encyclopedia of Genes and Genomes (KEGG) categorization. The *Y*-axis indicates the category, the *X*-axis the number of transcripts in a category. (**B**) Bubble chart of functional annotation of 4-fold differentially expressed genes based on KEGG categorization. The *Y*-axis indicates the signaling pathway category, the *X*-axis the enrichment factor. (**C**) Heat map of focused genes raw expression in transcriptome database. (**D**–**F**) Verification of gene expressions. Transcriptome sample CIK cells which were stably expressed with empty vector (CIK-vector) of VP4 (CIK-VP4) were seeded into 12-well plates for 24 h and examined for RLR related genes (**D**), NF-κB related genes (**E**) and TLR related genes (**F**) left). (**F**) right) VP4 promotes MyD88 production. CIK cells transiently transfected with VP4/empty vector were seeded in 6-well plates. Twenty-four hours later, cell lysate was used for WB analysis using MyD88 polyclonal antiserum and β-Tubulin as control. All the experiments independently were repeated at least three times. The histogram on the right exhibits the relative protein expression levels, which were quantified using ImageJ software. Data of qPCR are shown as mean ± SD of 4 wells of cell per group and are from one experiment representative of three independent experiments. Significance was calculated in relation to the control group. * *p* < 0.05, ** *p* < 0.01 (two tailed Student’s tests). The relative transcription levels were normalized to the transcription level of EF1α gene and are represented as fold induction relative to the transcription level in control cells, which was set to 1.

**Figure 6 biomolecules-10-00560-f006:**
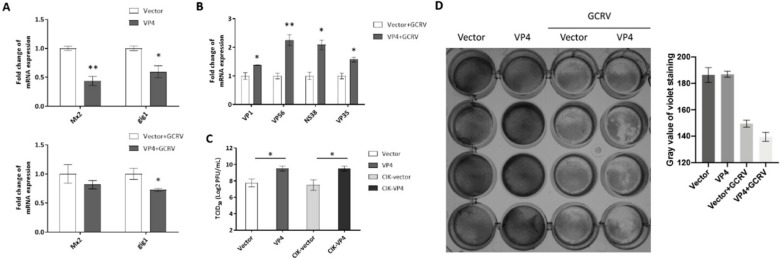
VP4 facilitated GCRV replication. (**A**) VP4 overexpression dampened ISGs mRNA expression in uninfected (upper) or GCRV infected (below) CIK cells. CIK cells transiently transfected with VP4/empty vector were seeded in 12-well plates. After 24 h, CIK cells were uninfected (above) or infected with GCRV (below). At 24 h post-infection, total RNA was extracted and examined for Mx2 and gig1 mRNA expression. (**B**) VP4 facilitated viral segments’ mRNA levels. CIK cells transiently transfected with VP4/empty vector were seeded in 12-well plates. After 24 h, CIK cells were infected with GCRV. At 24 h post-infection, total RNA was extracted and examined for VP1, VP56, NS38, and VP35 mRNA expression. (**C**) VP4 promotes GCRV infection. CIK cells were transiently transfected with VP4/empty vector as well as vector/VP4 stably transfected CIK cells (CIK-vector/CIK-VP4), were seeded in 6-well plates overnight and infected with GCRV, and the supernatants were collected at 24 h post-infection for viral titer assays by 50% tissue culture infective dose (TCID_50_). (**D**) VP4 accelerates GCRV-induced cell death. CIK cells transiently transfected with VP4/empty vector were seeded in 24-well plates. Twenty-four hours later, CIK cells were treated with PBS or infected with GCRV. 24 h post-infection, CIK cells were fixed and stained with crystal violet. All the experiments were repeated at least three times. The histogram exhibits the relative protein expression levels, which were quantified using ImageJ software. Data of qPCR and TCID_50_ are shown as mean ± SD of 4 wells of cell per group and are from one experiment representative of three independent experiments. Significance was calculated in relation to the control group. * *p* < 0.05, ** *p* < 0.01 (two tailed Student’s tests). The relative transcription levels were normalized to the transcription level of EF1α gene and are represented as fold induction relative to the transcription level in control cells, which was set to 1.

**Figure 7 biomolecules-10-00560-f007:**
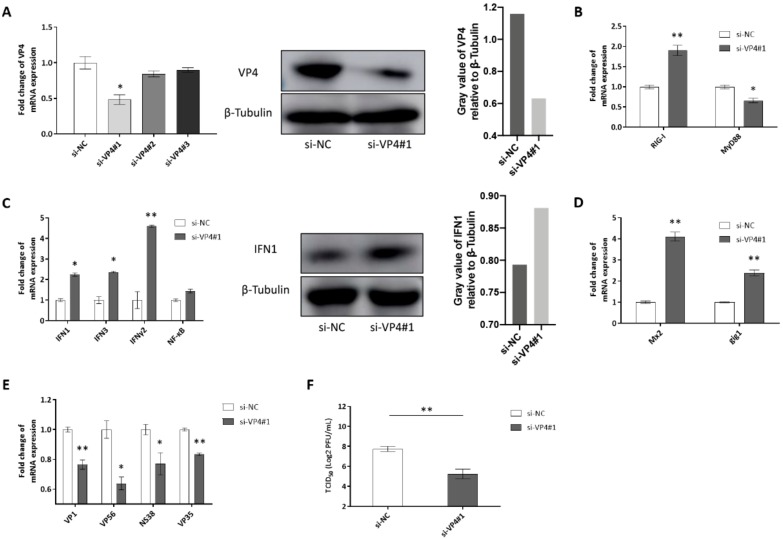
Effects of siRNA-mediated knockdown of VP4. (**A**) Left: Effects of siRNA on expression of GCRV VP4. CIK cells were seeded into 6-well plates overnight and transfected with 50 nM si-NC, si-VP4#1, si-VP4#2, or si-VP4#3. At 24 h post-transfection, the cells were infected with GCRV. At 24 h post-infection, total RNA was extracted to examine the transcriptional levels of VP4. Middle: Effect of siRNA#1 on protein expression of GCRV VP4. CIK cells were seeded into 6-well plates and transfected with 50 nM si-NC or si-VP41#1 and VP4-eGFP. At 24 h post-transfection, the cells were infected with GCRV. At 24 h post-infection, VP4 protein expression was detected with anti-GFP Ab. The experiment was repeated at least three times. Right: The histogram exhibits the relative protein expression levels, which were quantified using ImageJ software. (**B**–**E**) Effects of siRNA on PRRs (**B**), IFNs (C Left), ISGs (D) and the viral segment transcripts of GCRV (VP4 associated with GRP78 leading to unfolded protein response and ER stress VP4 associated with GRP78 leading to unfolded protein response and ER stress. (**E**) CIK cells were seeded into 6-well plates and transfected with 50 nM si-NC/si-VP41#1. At 24 h post-transfection, cells were infected with GCRV for 24 h before RT-qPCR. (**C**) Middle: CIK cells were seeded into 6-well plates overnight and transfected with si-NC/si-VP41#1. After GCRV infection for 24 h, cell lysate was IB with IFN1 polyclonal antiserum and β-Tubulin as control. Right: The histogram exhibits the relative protein expression levels, which were quantified using ImageJ software. (**F**) VP4 knockdown reduces GCRV infection. CIK cells seeded in 6-well plates overnight were infected with GCRV for 24 h and transiently transfected with si-NC or si-VP4#1. The supernatants were collected at 24 h post-transfection for viral titer assays by 50% tissue culture infective dose (TCID_50_). Data of qPCR and TCID_50_ are shown as mean ± SD of 4 wells of cell per group and are from one experiment representative of three independent experiments. Significance was calculated in relation to the control group. * *p* < 0.05, ** *p* < 0.01 (two tailed Student’s tests). The relative transcription levels were normalized to the transcription level of EF1α gene and are represented as fold induction relative to the transcription level in control cells, which was set to 1.

**Figure 8 biomolecules-10-00560-f008:**
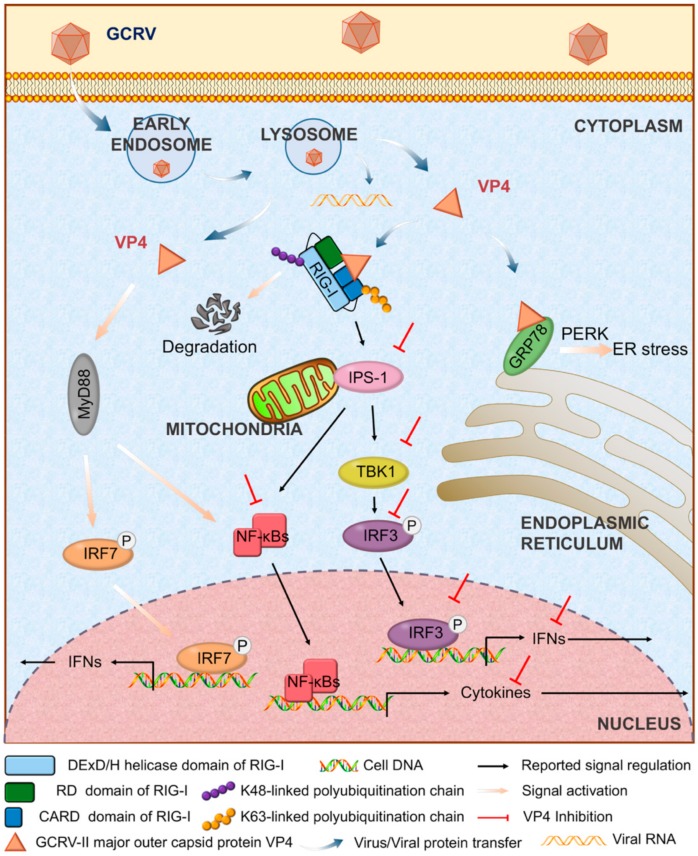
Regulatory model of VP4 in antiviral signaling in grass carp. Upon GCRV infection, GCRV-II major outer capsid VP4 enters into cytoplasm via early endosome and lysosome. VP4 targets RIG-I and GRP78 and results in different responses to suppress antiviral immunity and facilitate GCRV replication. VP4 interacts with CARD domain and RD domain of RIG-I in cytoplasm. After association with VP4, K48-linked ubiquitination of RIG-I is enhanced leading to RIG-I degradation. Consequently, signal transductions from RIG-I to downstream adaptors IPS-1 and STING are inhibited. Furthermore, VP4 restrains expression and phosphorylation of IRF3 and the subsequent signals of IFNs and NF-κB. VP4 binds GRP78 to activate ER stress via PERK-eIF2α mediated unfolded protein response. Meanwhile, dissociative VP4 protein also triggers MyD88-dependent TLR signaling pathway and induces production and phosphorylation of IRF7 and subsequent IFN secretion.

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
