# Peer review of "Grass Carp Reovirus Major Outer Capsid Protein VP4 Interacts with RNA Sensor RIG-I to Suppress Interferon Response"

_biomolecules, 2020, doi:10.3390/biom10040560_

Round 1
Reviewer 1 Report
The manuscript entitled “Grass carp reovirus major outer capsid protein vp4 interacts with RNA sensor RIG-I to suppresses interferon response” performed by Hang Su and colleagues describes the interaction between GCRV major outer capsid protein VP4 and RIG-I and how this interaction restrains interferon response and contributes to GCRV invasion.
This manuscript is a great contribution to the field of aquaculture and antivirals. The manuscript is interesting, well written, organized and well performed.
However the manuscript needs a minor revision before being accepted for publication:
- Line 115-117 / 642-643 paragraphs are too general, please further specify.
- Line 355: blocked or decreased? Please change or explain better.
- Line 356: suppressed or decreased? Please change or explain better.
- Line 359: Figure 4b does not show NFKB expression
- The image quality of some graphs is not well (Fig 4 and 5). Please improve them.
- In the fig 4C, in line 352, “the WB results illustrated that protein expression and phosphorylation of IRF3 were reduced by VP4 during GCRV infection”. The WB does not show that observation. Please substitute by a more representative image.
- The result in figure4H is confusing and not well explained in material and methods. In addition authors treated cells with MG132 and it is not explained why and it is not included in material and methods too.
- Please include statistical analysis for results in figure 4C,H,I, fig 5F right and fig 7A.
- Line 578: block or reduces? Please better explain
- Line 612: After VP4 stimulation, IRF7 was upregulated at mRNA and promoter activity level. IRF7 triggers IFN type 1 pathway and activates ISGS effectors. Could you please further explain why despite IRF7 upregulation there is not increase of IFN or ISGs levels?
Reviewer 2 Report
This is a very comprehensive study focusing on functional analysis of the interaction between a viral protein (VP4 of Grass carp reovirus) and RIG-I – an innate sensor of viral RNA. Several approaches are used to support the conclusions. F.ex, overexpression of the VP4 protein is followed by VP4 knock out studies to support the data. Research on teleost virology and immunology is lacking functional studies like this and it is an important contribution to this field of research.
Manuscript title: “suppresses” should be “suppress”
Introduction
Line 44/45 – reasoning behind these statements is not clear to me – why should you investigate the function of viral proteins? Modify. Is it necessary to include the first paragraph at all (lines 35-45).
Lines 64-84: It is not so clear from the text which parts of the RIG-I pathway is actually functionally characterized in fish. Please re-write/clarify.
Lines 126-129: It is not clear from the text or the referring literature how the specificity of some of the abs was tested (anti IFN1, anti IRF7, anti Myd88, anti VP4 Line 156). Has to be included.
Line 344: What is ment by “limited antibodies of fish”? Does it mean there are limited number of antibodies available to relevant markers? Please re-write.
Lines 363-364: The sentence is lacking referral to Fig 4 I.
Line 419: “were screened by G418” – more appropriate to write “were selected by G418 treatment”? If I have understood this correctly.
Line 435 -437: “This result demonstrated that VP4 activated the MyD88-dependent TLR signaling pathway during GCRV 436 infection”. To demonstrate actual activation of Myd88 additional data is needed (more data than increased Myd88 expression). Please rephrase statement.
Line 465: “Crystal violet staining also showed that VP4 promotes the cell death caused by GCRV infection (Fig. 6D)” . The difference in staining between vector+ GCRV and VP4+GCRV is small and maybe not significant? The phrasing should be changed from “show” to “indicates”.
Line 492: Is this the heading of the paragraph?
Line 541: Phrasing is not clear – simplify/re-write.
Line 566: Phrasing is not clear – re-write.
Line 582: “Whether upon” not clear to me what this means. Please rephrase the sentence.
Line 583/84: Usually homology between teleost and human type I IFNs is low – is this statement correct?
Line 599: “GCRV replication lost limitation” – what is does this mean? Re-phrase.
Reviewer 3 Report
The manuscript entitled: “Grass carp reovirus major outer capsid protein VP4 interacts with RNA sensor RIG-I to suppresses interferon response“ is considered for publication in Biomolecules. Authors investigated the impact of GCRV major outer capsid protein VP4 on type I IFN responses by interaction and disruption of foreign RNA sensing by RIG-I receptor. The used an impressive array of methods: GST pull-down, immunoprecipitation, LC-MS/MS, western blotting, confocal and transmission electron microscopy, far-red mNeptune-based bimolecular fluorescence complementation system, luciferase reporter assay, RNAseq transcriptome analysis and siRNA mediated knockdown. Using this methods Authors were able to show that VP4 is master manipulator of the antiviral response of virus infected cells. Virus is using the major outer capsid to bind the PRRs and limit signalling which could lead to induction the type I IFN response. The methods are used correctly, and the paper is well written. I like the summary figure 8 – is there possibility that this figure is transformed into “graphical abstract”?. I have only couple minor comments and requests which could be addressed before publication:
Line 38: However, diseases…
Line 247: what transfection agent was used for siRNA? This information should be added to in this line.
Line 262: Statistical analysis description should be added to the material and methods
Figure 5A: might need higher resolution
Figure 7B and 7D: is the whole section present ?– it seems that something was cut away during the PDF formatting
